# Expression Profile of Stemness Markers CD138, Nestin and Alpha-SMA in Ameloblastic Tumours

**DOI:** 10.3390/ijerph18083899

**Published:** 2021-04-08

**Authors:** Callisthenis Yiannis, Massimo Mascolo, Michele Davide Mignogna, Silvia Varricchio, Valentina Natella, Gaetano De Rosa, Roberto Lo Giudice, Cosimo Galletti, Rita Paolini, Antonio Celentano

**Affiliations:** 1Melbourne Dental School, University of Melbourne, 720 Swanston Street, Carton, VIC 3053, Australia; c.yiannis@student.unimelb.edu.au (C.Y.); rita.paolini@unimelb.edu.au (R.P.); 2Pathology Unit, Department of Advanced Biomedical Sciences, University Federico II of Naples, 80131 Naples, Italy; massimo.mascolo@unina.it (M.M.); silvia.varricchio@gmail.com (S.V.); valenatella@gmail.com (V.N.); gaderosa@unina.it (G.D.R.); 3Department of Neurosciences, Reproductive and Odontostomatological Sciences, University Federico II of Naples, 80131 Naples, Italy; mignogna@unina.it; 4Department of Clinical and Experimental Medicine, University of Messina, 98122 Messina, Italy; roberto.logiudice@unime.it; 5Comprehensive Dentistry Department, Faculty of Dentistry, Universitat de Barcelona, L’Hospitalet de Llobregat, 08007 Barcelona, Spain; cosimo88a@gmail.com

**Keywords:** ameloblastoma, ameloblastic carcinoma, nestin, CD138, syndecan-1, alpha-SMA, stemness markers

## Abstract

Ameloblastic carcinoma is a rare malignant odontogenic neoplasm with a poor prognosis. It can arise de novo or from a pre-existing ameloblastoma. Research into stemness marker expression in ameloblastic tumours is lacking. This study aimed to explore the immunohistochemical expression of stemness markers nestin, CD138, and alpha-smooth muscle actin (alpha-SMA) for the characterisation of ameloblastic tumours. Six cases of ameloblastoma and four cases of ameloblastic carcinoma were assessed, including one case of ameloblastic carcinoma arising from desmoplastic ameloblastoma. In all tumour samples, CD138 was positive, whilst alpha-SMA was negative. Nestin was negative in all but one tumour sample. Conversely, the presence or absence of these markers varied in stroma samples. Nestin was observed in one ameloblastic carcinoma stroma sample, whilst CD138 was positive in one ameloblastoma case, one desmoplastic ameloblastoma case, and in two ameloblastic carcinoma stroma samples. Finally, alpha-SMA was found positive only in the desmoplastic ameloblastoma stroma sample. Our results suggest nestin expression to be an indicator for ameloblastic carcinoma, and CD138 and alpha-SMA to be promising biomarkers for the malignant transformation of ameloblastoma. Our data showed that nestin, CD138, and alpha-SMA are novel biomarkers for a better understanding of the origins and behaviour of ameloblastic tumours.

## 1. Introduction

Ameloblastomas (ABs) are uncommon, benign, locally aggressive odontogenic tumours of epithelial origin with a high incidence of recurrence [1,2,3,4]. Left untreated, they have the potential to reach large sizes and cause physical disfiguration and functional disturbances. There is no gender preference, but there is a high incidence in the third and fourth decades of life [4,5]. Ameloblastic tumours show significant histological variations and are classified into various benign and malignant entities [1]. According to the WHO 2017 classification, benign ABs are categorised into conventional, unicystic, and extraosseous/peripheral types [6]. Conventional is the most common type and makes up 85% of cases [4,7]. Histologically, it can be categorised into follicular and plexiform [1,2]. Other less common histological variants are clear cell and desmoplastic cell [8]. Desmoplastic ABs behave like conventional ABs, although their clinical and radiographic characteristics may be different [8,9,10].

In some cases, ABs can demonstrate metastasis with benign histological features. This type is classified as a metastasising (malignant) AB. This was originally classified as a malignant form in the 2005 WHO classification system, but has since been re-classified as a benign epithelial odontogenic tumour in the current 2017 WHO classification [6]. Ameloblastic carcinoma (AC) is an AB that can demonstrate metastatic histological features and malignant cytological characteristics. Accordingly, it is categorised as a malignant odontogenic tumour under the 2017 WHO classification [6]. Genetic and molecular alterations in these odontogenic epithelial tumours have been identified as possible associations with mechanisms of oncogenesis, cyto-differentiation, and tumour progression [11,12]. The development of an AC from an existing AB is extremely rare. Only 16 cases have been reported in literature in the last 10 years [13]. AC is generally characterised by high morbidity and mortality, and the survival rates of patients diagnosed with AC are significantly reduced in those with evidence of metastasis [14,15].

Stem cells are undifferentiated cells capable of self-renewal and the production of a diverse range of differentiated cells [16]. Tumours can contain a heterogenous population of stem cells known as cancer stem cells (CSCs). CSCs have the characteristics of self-renewal driving tumorigenesis [16,17]. Studies have suggested that haematopoietic and neural stem cell markers may play important roles in epithelial–mesenchymal interactions and cell proliferation/differentiation in both odontogenic epithelial tumours and during odontogenesis [18,19,20,21]. Specifically, several immunohistochemical studies have outlined an effective correlation between the levels of some markers of stemness such as nestin, CD138, and alpha-smooth muscle actin (alpha-SMA) and the different forms of ameloblastic tumours [22].

Nestin is an intermediate filament constituting the cytoskeleton, and is a marker of neural stem cells or progenitor cells [23]. The expression of nestin is related to tooth development and repair of dentine [24]. The expression of nestin in ameloblastomas and malignant ameloblastomas has been reported in the literature to be negative [24]. CD138, also known as syndecan-1, is a peptide that inhibits tumour growth. It is highly expressed in fibroblasts and epithelial cells [25]. Alterations in CD138 expression results in changes to cell behaviour, shape, growth, migration, and cytoskeletal organisation [26]. To date, CD138 expression and its role in AC remains debated. Alpha-SMA, a marker of myofibroblasts, has also been reported in studies comparing ABs to ACs [27]. A positive correlation has been identified between the number of myofibroblasts present in the stroma and the aggressive behaviour of odontogenic tumours through enhancement of epithelial–mesenchymal interactions [28,29]. Research has demonstrated that the clinical recurrence of ABs may be predicted by alpha-SMA expression [30]. CD138 and alpha-SMA expression may indicate a higher aggressive potential of AB [31], and alpha-SMA expression in epithelial cells may indicate AC [27,31,32].

Cases presenting with subtle metastatic change, atypical phenotypes, or poor biopsied tissue reduce the ability of an accurate diagnosis. The understanding of the role of stemness markers in ABs may aid in the early diagnosis of malignant ameloblastic tumours with direct implications on their management protocol [33].

Research into stemness marker expression in ameloblastic tumours is currently deficient and evidence is controversial. Further studies are required to better understand their role in this type of malignancy. The main aim of this study was to explore the immunoexpression of stem cell markers nestin, CD138 (syndecan-1), and alpha-SMA in a series of four cases of ameloblastic carcinoma and to compare this data to an ameloblastoma immunoexpression profile.

Our results suggest nestin expression to be an indicator of AC, and CD138 and alpha-SMA to be biomarkers for transformation of AC from AB. Our study confirms the role of nestin, CD138 (syndecan-1), and alpha-SMA as promising biomarkers for a better understanding of the origin and behaviour of ameloblastic tumours.

## 2. Materials and Methods

This study was approved by the Ethics Committee of the University “Federico II” of Naples, Italy (protocol n. 35/15). Appropriate permission and written informed consent were obtained from all the patients described in this article.

For this study, we selected a case of desmoplastic AB (female, 22 years of age) that progressed to AC, three further ACs (two males and a female; mean age: 57 years; age range: 48–73 years), and six ABs from the archives of the Pathology Section of Advanced Biomedical Sciences at the Federico II University of Naples, Italy; every patient had a follow-up of at least 24 months.

Each specimen was fixed in 10% buffered formalin, embedded in paraffin, and serially sectioned (4 μm thick sections). For each case, one section was stained with haematoxylin and eosin (H&E) and the others used for immunohistochemistry (labelled streptavidin-biotin standard technique) with anti-nestin (nestin, 10c2, Santa Cruz Biotechnology, Santa Cruz, CA, USA, diluted 1:100), anti-CD138/syndecan-1 (CD138, B-A38, Ventana, Tucson, AZ, USA, prediluted), and anti-α-smooth muscle actin (αSMA, 1A4, Ventana, Tucson, AZ, USA, prediluted) antibodies. Cells showing definite brown staining were judged positive for nestin, CD138, and alpha-SMA. All slides were examined in a double-blinded fashion by two pathologists (M.M. and G.D.R.) to confirm the diagnosis and to assess the immunohistochemical staining, both in tumour islands and stromal fibroblasts, according to a semiquantitative score, as negative, focal (+), moderate (++), and extensive (+++) positivity.

## 3. Results

Four cases of AC and six cases of AB were included in this study. Of these six, one case of desmoplastic AB progressed to AC. AC case descriptions can be found below.

Case 7: A 22-year-old female presented with a four-month history of an asymptomatic expansive lesion (bucco-lingually) involving quadrant 4, region 41–46 (Figure 1a–c). Significant loss of soft tissue and bony attachment (up to 100%) was evident. Increased vasculature and agenesis of the mandibular second premolars was observed (Figure 1a–c). The patient was a non-smoker and non-drinker. Diagnosis at this stage was a desmoplastic AB and treated with box resection including removal of 41–44 and 85. The one month follow-up indicated healthy and healing tissues (Figure 1d,e). The seven-month follow-up was characterised by the onset of an erythematous lesion on the buccal gingiva in the area of 31 (Figure 1f,g) and radiographic findings were also suggestive of a potential recurrence. Diagnosis at this stage was an AC and treated with a more extensive box resection including removal of 31 (Figure 1h). The histopathological description of this case is presented in Figure 2.

Case 8: A 48-year-old male presented with a history of a five-year lesion. Upon examination, the lesion was a 4 cm × 7 cm oral soft tissue ulceration with recurrent abscess and pus of the right mandible. Multiple teeth had been lost and swelling of the submandibular and lymph nodes of the neck were present homolaterally. The patient was a smoker (17 cigs/day) and a casual drinker. Treatment involved hemi-mandibulectomy. The histopathological description is presented in Figure 3a–d.

Case 9: A 73-year-old male presented with a solid tissue lesion involving destruction of the roof and anterior wall of the maxillary sinus infiltrating the nasal cavity and hard palate. The patient had a previous history (four years prior) of a basal cell carcinoma (BCC) and treatment involved excision of the entire right orbit. One year following treatment, the patient was diagnosed with a BCC of the right nose wing that was excised. The patient had hypertension and was a smoker (60 cigs/day). The histopathological description is presented in Figure 3e–h.

Case 10: A 50-year-old female presented with a carcinoma (33 mm in diameter) involving the right nasal fossa, ethmoidal bone, maxillary bone, maxillary sinus, and floor of the orbit. The patient was a non-smoker and non-drinker. The patient received maxillectomy and reconstruction with titanium mesh. The histopathological description is presented in Figure 3i–l.

All histopathologic diagnoses of AB and AC were confirmed. ABs were typically composed of epithelial nests of columnar cells arranged in a palisading pattern surrounding a loose network of cells mimicking a stellate reticulum, in a loose connective stroma. ACs showed the typical features of malignancy, including marked nuclear atypia, a high mitotic index, and neural or vascular invasion, in the context of classical AB.

All ABs expressed CD138, with a variable degree of expression varying from + to +++, in their solid tumour counterpart; the stromal fibroblasts resulted negative, except for a case showing a moderate positivity (++) for CD138.

Nestin was negative in all AB cases. Alpha-SMA was negative in all AB cases except for one case (positive only in the stromal component).

The AC derived from the desmoplastic AB expressed alpha-SMA focally (+) in the smooth muscle around some tumour nests; CD138 diffusely stained the tumour islands (+++) and moderately (++) the stroma. Nestin was negative.

The three remaining AC cases were negative for alpha-SMA. Only one case showed an extensive (+++) positivity for nestin, both in the tumour cells and stromal fibroblasts.

Finally, one case of AC resulted positive for CD138, both in the tumour cells (++) and fibroblasts (+++), one case showed an extensive (+++) positivity only in the tumour cells, and one case was completely negative.

Table 1 summarises the immunostaining data.

Statistical analysis of these stemness markers did not show significant differences (*p* < 0.05; Fisher’s exact test). This was expected due to the rarity of ACs and the consequent sample size in our cohort.

## 4. Discussion

ABs are uncommon, benign, locally aggressive odontogenic tumours of epithelial origin with a high incidence of recurrence [1,2,3,4] and potential for malignant transformation into a metastasising (malignant) AB or AC [15,34]. To date, there are approximately 65 cases of metastasising (malignant) AB and 125 cases of AC reported in the literature [15,34]. ABs present radiographically as uni- or multilocular radiolucencies frequently with cortical expansion. Clinicians should suspect a malignant lesion when there are presentations of paraesthesia, pain, irregular borders, and invasion of adjacent tissues [14,35].

ACs usually present with microscopic evidence of malignancy. However, confirmation of whether this is a secondary AC must be acknowledged by a history of persistent, recurrent or residual AB [36]. Differential diagnosis between the lesions, from incisional biopsies, can prove to be difficult. Differential diagnosis often includes other types of intra-osseous carcinomas of the jaws [36]. Immunohistochemistry can be utilised to aid in the diagnosis and classification of odontogenic tumours. Some currently published immunohistochemical markers used to differentiate AC from AB include CK18, parenchymal MMP-2, stromal MMP-9, Ki-67, and p53 [36,37]. Novel immunomarkers are crucial for a better understanding of lesion origins, diagnosis, and behaviour.

Diagnostic difficulty is sometimes encountered with ABs of unusually aggressive behaviour and in differentiation from ACs. To identify the markers of stemness that will have implications in the diagnosis of AC and cases with subtle malignant transformation, immunohistochemical expression of a panel of markers (nestin, CD138, and alpha-SMA) was investigated.

Nestin is a CSC surface marker identified in tissues and pathological conditions, such as the neural crest, heart, testis, reactive astrocytes (after brain injury), and the central and peripheral nervous systems [24]. Previous studies have discussed the expression of nestin as a useful marker for the identification of odontogenic ectomesenchyme and odontoblasts in odontogenic tumours [24,38]. Previously published data by Fujita et al. (2006) indicated that almost all of their cases of ameloblastomas and malignant ameloblastomas (three cases) were negative for nestin [24]. This was reflected in our experimental study. However, amongst our cases of AC, 25% of our cohort had an extensive expression of nestin in both the tumour islands and stromal cells. Analogous to the literature, there was no expression of nestin in any of our AB cases. Although our sample size was small and further studies should aim to better elicit its expression in larger cohorts, nestin appears to have immunoreactivity in malignant ameloblastic tumours.

CD138 is highly expressed in fibroblasts and epithelial cells and functions to inhibit tumour growth [38]. The expression of CD138 in ABs and ACs is controversial amongst the literature. Various studies have reported different findings in regard to increased or decreased expression in ACs. A study published by Bologna-Molina et al. (2009) suggested CD138 expression in desmoplastic AC to be inversely correlated to the proliferative index Ki67 [39]. Therefore, according to these authors, decreased CD138 expression in desmoplastic AB corresponds with its higher aggressiveness [39]. However, a study published by Martínez-Martínez et al. (2017) indicated CD138 is mainly expressed in the peripheral cells of ABs and is expressed in most areas of AC [36]. The AB cases in our cohort indicated the expression of CD138 (from focal to extensive) in the solid tumour but negative in the stellate tumour areas and stromal fibroblasts. The case of desmoplastic AB resulted positive in fibroblasts and tumour islands. A particular focus of our study was to assess the progression of AC from a desmoplastic AB. The AC arising from the desmoplastic AB showed CD138 to have greater positivity in the tumour islands but less positivity in the stroma. All cases of AC were positive for CD138 in the tumour, and 50% of cases were positive in the stroma. Our data suggest that ameloblastic tumours may be positive for CD138 in tumour islands. This is in accordance with research published by Martínez-Martínez [36]. Furthermore, developing AC from desmoplastic ABs may be positive for CD138 in tumour islands and stromal cells. Further studies should aim to confirm these results as a biomarker for transformation of AC from AB.

Alpha-SMA, a marker of myofibroblasts, has been assessed in the literature for its potential to determine the recurrence of AB and as a biomarker of transformation to AC [27,30,31,32]. Interestingly, in our experimental study, only the case of desmoplastic AB and the developing AC resulted positive for alpha-SMA. This was only in the surrounding tumour nests. All other cases of AB and AC were negative for alpha-SMA. A study published by Siar and Ng (2019) assessed the epithelial–mesenchymal transition of neoplastic cells, as it is essential for metastatic expansion and cancer progression [40]. It was concluded that stromal upregulation of alpha-SMA (as well as osteonectin and N-cadherin) implicates a role in local invasiveness [40]. Further research suggests a positive correlation between the number of myofibroblasts present in the stroma and the aggressive behaviour of odontogenic tumours [28]. This suggests a histopathological feature for a developing AC. Considering alpha-SMA expression in epithelial cells may indicate AC [27,31,32,41], further studies should aim to assess alpha-SMA expression in fibroblasts and epithelial cells.

## 5. Conclusions

The present study had the opportunity to assess the rare progression of AC from a desmoplastic AB. The immunohistochemical results suggested nestin expression to be an indicator for AC, and CD138 and alpha-SMA to be biomarkers for transformation of AC from AB. However, further experiments are required to look at tumour and stroma tissue in order to elucidate the presence or absence of these biomarkers. This will allow a better understanding of the expression of stemness markers amongst a larger cohort of cases and progressively shed light on their immunoexpression amongst ameloblastic tumours.

## Figures and Tables

**Figure 1 ijerph-18-03899-f001:**
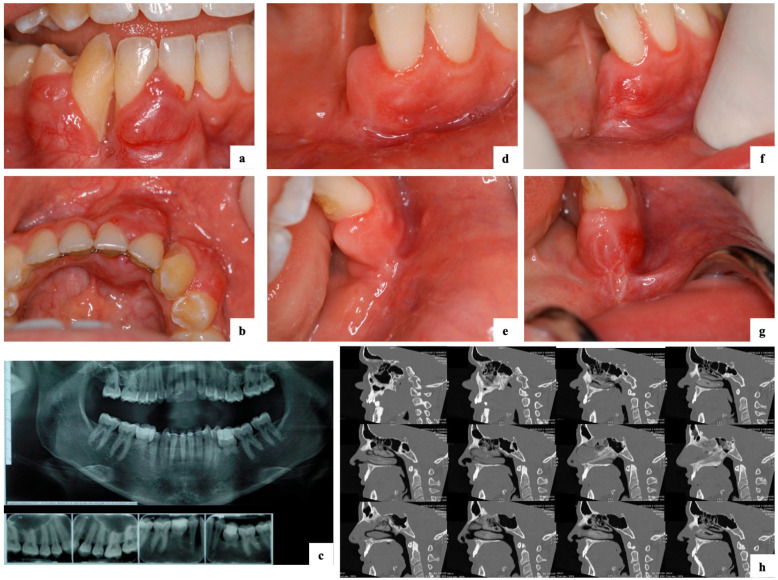
Initial presentation of a 22-year-old female with a desmoplastic ameloblastoma that progressed to an ameloblastic carcinoma (case 7). (**a**–**c**) Presentation with a four-month history of an asymptomatic expansive lesion (bucco-lingually) involving quadrant 4, region 41–46, with significant loss of soft tissue and bony attachment (up to 100%). Orthopantomogram (OPG) and peri-apical radiographs indicated agenesis of mandibular second premolars and retention of 75 and 85. Histopathologic diagnosis at this stage was a desmoplastic ameloblastoma. (**d**,**e**) Clinical presentation indicated healthy and healing tissues one-month after surgical intervention via box resection and removal of 41–44 and 85. (**f**,**g**) Clinical presentation at the seven-month follow-up indicated an erythematous lesion on the buccal gingiva in the area of 31. Diagnosis was an ameloblastic carcinoma and treated with a more extensive box resection including removal of 31. (**h**) Computed tomography (CT) scan one-month after surgery indicated removal of the ameloblastic carcinoma.

**Figure 2 ijerph-18-03899-f002:**
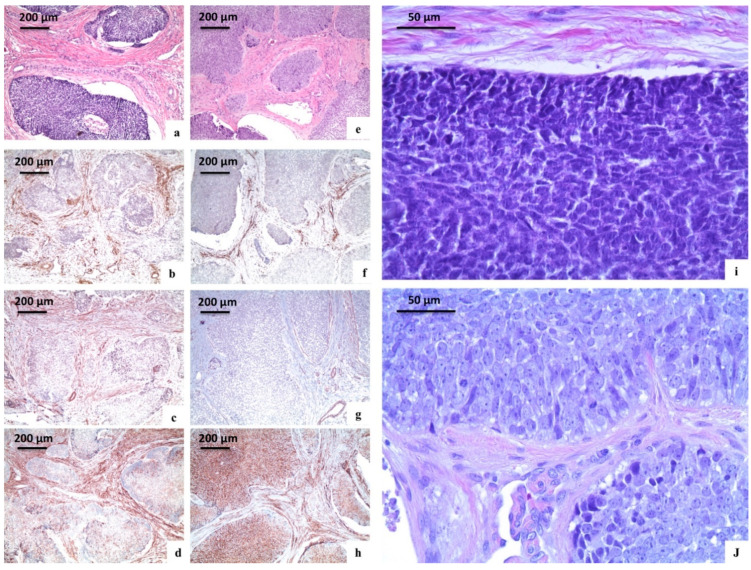
Immunohistochemical analysis of the stemness markers (nestin, CD138, and alpha-SMA) in an ameloblastic tumours. (**a**–**d**,**i**) Case 6; (**e**–**h**,**j**) case 7. (**a**) Follicular ameloblastoma (haematoxylin and eosin, original magnification ×100); (**b)** staining for nestin in follicular ameloblastoma (original magnification ×100); (**c**) staining for alpha-SMA in follicular ameloblastoma (original magnification ×100); (**d**) staining for CD138 in follicular ameloblastoma (original magnification ×100); (**e**) evolving desmoplastic ameloblastoma (haematoxylin and eosin, original magnification ×100); (**f**) staining for nestin in evolving desmoplastic ameloblastoma (original magnification ×100); (**g**) staining for alpha-SMA in evolving desmoplastic ameloblastoma (original magnification ×100); (**h**) staining for CD138 in evolving desmoplastic ameloblastoma (original magnification ×100); (**i**) follicular ameloblastoma: High magnification (haematoxylin and eosin, original magnification ×400); (**j**) evolving desmoplastic ameloblastoma: High magnification (haematoxylin and eosin, original magnification ×400).

**Figure 3 ijerph-18-03899-f003:**
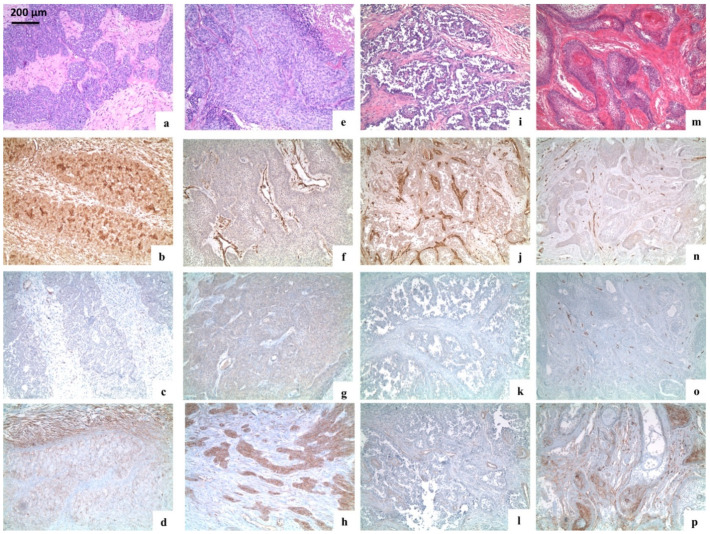
Immunohistochemical analysis of the stemness markers (nestin, CD138, and alpha-SMA) in ameloblastic tumours. (**a**–**d**) Case 8; (**e**–**h**) case 9; (**i**–**l**) case 10; (**m**–**p**) case 6. (**a**,**e**,**i**) Ameloblastic carcinomas (haematoxylin and eosin, original magnification ×100); (**m**) desmoplastic ameloblastoma (haematoxylin and eosin, original magnification ×100); (**b**,**f**,**j**) staining for nestin in ameloblastic carcinomas: Overexpression of nestin was seen in tumour cells, fibroblasts, endothelial cells, and lymphocytes (**b**: original magnification ×100); (**f**,**j**) tumour cells and fibroblasts were negative, while endothelial cells were positive (original magnification ×100); (**n**) staining for nestin in recurrent plexiform ameloblastoma: Tumour cells and fibroblasts were negative (original magnification ×100); (**c**,**g**,**k**) staining for alpha-SMA in ameloblastic carcinoma: Alpha-SMA was negative; a weak expression of alpha-SMA was present in the vessel wall (**c**,**g**,**k**: original magnification ×100); (**o**) staining for alpha-SMA in recurrent plexiform ameloblastoma: Tumour cells and fibroblasts were negative, but a weak expression was present in the vessel wall (original magnification ×100); (**d**,**h**,**l**) staining for CD138 in ameloblastic carcinoma: (**d**) Tumour cells and stromal fibroblasts stained an extensive and moderate expression of CD138, respectively (**d**: original magnification ×100); (**h**) overexpression of CD138 was seen only in tumour islands (**h**: original magnification ×100); (**l**) only a minority of tumour cells resulted positive for CD138 (**l**: original magnification ×100); (**p**) immunostaining for CD138 in recurrent plexiform ameloblastoma: Tumour islands were variably positive for CD138.

**Table 1 ijerph-18-03899-t001:** Immunoexpression profile of all stemness markers (nestin, CD138, and alpha-smooth muscle actin (alpha-SMA)) in ameloblastic tumours.

Case n.	Diagnosis	Nestin		CD138 (Syndecan-1)		Alpha-SMA	
		Tumour	Stroma	Tumour	Stroma	Tumour	Stroma
1	Ameloblastoma	-	-	++	-	-	-
2	Ameloblastoma	-	-	+	-	-	-
3	Ameloblastoma	-	-	+++	-	-	-
4	Ameloblastoma	-	-	+	-	-	-
5	Ameloblastoma	-	-	++	-	-	-
6	Desmoplastic ameloblastoma	-	-	++	+++	-	++
7	Ameloblastic carcinoma derived from desmoplastic ameloblastoma	-	-	+++	++	-	+
8	Ameloblastic carcinoma	+++	+++	++	+++	-	-
9	Ameloblastic carcinoma	-	-	+++	-	-	-
10	Ameloblastic carcinoma	-	-	+	-	-	-

## Data Availability

The data presented in this study are available on request from the corresponding author.

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
