# Peer review of "Expression Profile of Stemness Markers CD138, Nestin and Alpha-SMA in Ameloblastic Tumours"

_ijerph, 2021, doi:10.3390/ijerph18083899_

Round 1

Reviewer 1 Report

  1. The overall writing would benefit from some grammatical review.
  2.  The quantitative scoring system used for the IHC has to be described and defined in Material and Methods.
  3. Table 1 should be listed by cases.
  4. The quality of the images shown in Fig 1 are generally poor.
  5. Legends of Fig.1 and Fig. 2 should have a title. 
  6. Why authors are not showing an example of Ameloblastoma?
  7. To evaluate expression profile of stemness markers you would need to use other methodologies as well, not only IHC. For example, flow cytometry and RT-qPCR.

Author Response

Reviewer comments and author responses

Reviewer 1

Comment 1: The overall writing would benefit from some grammatical review.

Reply: Thank you for this comment. The paper has been extensively revised by two native English speakers.

Comment 2: The quantitative scoring system used for the IHC has to be described and defined in Material and Methods.

Reply: Thank you for this valuable comment. According to a suggestion made by reviewer 3, we have now changed our scoring system to a semiquantitative score, as negative, focal (+), moderate (++) and extensive (+++) positivity. The new scoring system used has been described in the materials and methods section of this draft.

Comment 3: Table 1 should be listed by cases.

Reply: Thank you for this valuable comment. We included the case list in table 1 as suggested.

Comment 4: The quality of the images shown in Fig 1 are generally poor.

Reply: Thank you for this valuable comment. We have added another figure to our manuscript, therefore figure 1 and 2 now read as figures 2 and 3. Following your suggestion, figures 2 and 3 have been re-processed by professional graphical designers to achieve top standard quality at a resolution of 300dpi (each figure).

Comment 5: Legends of Fig.1 and Fig. 2 should have a title.

Reply: Thank you for this valuable comment. We have updated the figure legends in the attached manuscript accordingly.

Comment 6: Why authors are not showing an example of Ameloblastoma?

Reply: Thank you for this suggestion. Following your suggestion, we decided to include both clinical and radiographic presentation of the most interesting case of our series (currently labelled as figure 1). We would like to highlight to this reviewer that two cases of ameloblastoma (histopathological description) were already presented in Figures 2a and 3m.

Comment 7: To evaluate expression profile of stemness markers you would need to use other methodologies as well, not only IHC. For example, flow cytometry and RT-qPCR.

Reply: Thank you for this valuable comment. We would have hoped to measure the expression of stemness markers using additional techniques such as ELISA and Western Blot. However, our study was performed on a retrospective database of FFPE tissues with no access to fresh samples of any sort. Flow cytometry would have been more useful with living cells. RT-qPCR would have been indicated to assess the coding genes for the markers of interest without any quantification of secreted proteins. Additionally, no allocated grants have been obtained for this project. Furthermore, supplementary more complex experiments are beyond the scope of the current paper.

Reviewer 2 Report

Dear authors,

The study is really interesting

Please, could the authors add scale bar in the images in the figures?

The authors should add a quantification in the the figures from the images

Best regards,

Juan

Author Response

Reviewer 2

Comment 1: Could the authors add scale bar in the images in the figures?

Reply: Thank you for the overall positive comments. We have added scale bars into the multi-panel figures.

Comment 2: The authors should add a quantification in the figures from the images

Reply: Thank you for this comment. Our understanding is that you have suggested we add a quantification of the staining in the multi-panel figures from the table data. Based on the previous comment we are very happy to insert a scale bar to the multi-panel figure. However, we are concerned that this further addition may over complicate and clutter the figures. Now that table 1 and all figure legends has been updated with case numbers, readers are able to easily cross reference data.

Reviewer 3 Report

1. A summary of pathological features of enrolled patients is suggested. 2. Relapse is one of the main characteristics of ameloblastoma after surgery. Did the author include the patient with recurrence? 3. It is understandable that ameloblastoma/ameloblastic carcinoma is very rare, but still it would be not that convincing to show the potential value of using CD133/CD138/alpha-SMA as novel biomarkers with less than 10 samples. 4. Table 1: is there any difference among all CD138 positive tumor samples? Can the scoring be further categorized as “+,++,+++” base on the expression profile?

Author Response

Reviewer 3

Comment 1: A summary of pathological features of enrolled patients is suggested.

Reply: Thank you for this valuable comment. We have included a detailed summary of the pathological features of the enrolled patients in the results section. Additionally, comprehensive iconography (clinical and radiographical findings) of the most interesting case of our series was added (figure 1).

Comment 2: Relapse is one of the main characteristics of ameloblastoma after surgery. Did the author include the patient with recurrence?

Reply: Thank you for this interesting comment. The original transcript includes and discusses the patient with desmoplastic ameloblastoma that progressed to an ameloblastic carcinoma. This has been outlined in the results, discussion and conclusion. Additionally, following your suggestion we emphasise the case that transformed from ameloblastoma to ameloblastic carcinoma by adding a detailed iconography with clinical and radiographical findings of this specific case. Now that table 1 and all figure legends has been updated with case numbers, readers are also able to easily cross reference data.

Comment 3: It is understandable that ameloblastoma/ameloblastic carcinoma is very rare, but still it would be not that convincing to show the potential value of using CD133/CD138/alpha-SMA as novel biomarkers with less than 10 samples.

Reply: Thank you for this valuable comment. Ameloblastic carcinoma is very rare, with less than 150 cases described globally to date. Keeping this in mind, we are potentially presenting here, one of the biggest case series in online literature for ameloblastic carcinoma as a single centre. Furthermore, the development of an ameloblastic carcinoma from an ameloblastoma is even more rare with only 16 cases reported in literature in the last 10 years. We acknowledge that validating novel biomarkers would require a larger sample size but given the rarity of disease, we are aiming with this contribution to the literature to foster future multicentre studies.

Comment 4: Table 1: is there any difference among all CD138 positive tumor samples? Can the scoring be further categorized as “+,++,+++” base on the expression profile?

Reply: Thank you for this inspirational comment. We have now changed our scoring system to a semiquantitative score, as negative, focal (+), moderate (++) and extensive (+++) positivity. All cases have been rescored accordingly by two blind pathologists. The new scoring system used has been described in detail in the materials and methods section of this draft.

Round 2

Reviewer 3 Report

No further comments.